# Molecular characterization of the permanent outer-inner membrane contact site of the mitochondrial genome segregation complex in trypanosomes

**Philip Stettler[1,2], Bernd Schimanski[1], Salome Aeschlimann[1], André Schneider ⬡[1] ***

**1** Department of Chemistry, Biochemistry and Pharmaceutical Sciences, University of Bern, Bern, Switzerland, **2** Graduate School for Cellular and Biomedical Sciences, University of Bern, Bern, Switzerland

\* andre.schneider@unibe.ch

**Data Availability Statement:** All relevant data are within the paper and its Supporting Information files.

## Abstract

The parasitic protozoan *Trypanosoma brucei* has a single unit mitochondrial genome linked to the basal body of the flagellum via the tripartite attachment complex (TAC). The TAC is crucial for mitochondrial genome segregation during cytokinesis. At the core of the TAC, the outer membrane protein TAC60 binds to the inner membrane protein p166, forming a permanent contact site between the two membranes. Although contact sites between mitochondrial membranes are common and serve various functions, their molecular architecture remains largely unknown. This study elucidates the interaction interface of the TAC60-p166 contact site. Using *in silico*, *in vitro*, and mutational *in vivo* analyses, we identified minimal binding segments between TAC60 and p166. The p166 binding site in TAC60 consists of a short kinked α-helix that interacts with the C-terminal α-helix of p166. Despite the presence of conserved charged residues in either protein, electrostatic interactions are not necessary for contact site formation. Instead, the TAC60-p166 interaction is driven by the hydrophobic effect, as converting conserved hydrophobic residues in either protein to hydrophilic amino acids disrupts the contact site.

## Author summary

Mitochondria are surrounded by two membranes and essential for nearly all eukaryotes. Contact sites between the two membranes are important for mitochondrial function. However, most contact sites are dynamic making their molecular architecture challenging to study. The tripartite attachment complex (TAC) of parasitic protozoan *Trypanosoma brucei* connects its compact mitochondrial genome with the basal body of the flagellum. This couples the segregation of the replicated mitochondrial genome to the old and new basal body. The TAC contains permanent contact sites formed by the outer membrane protein TAC60 and the intermembrane space-exposed C-terminus of p166 of the inner membrane. We have used it as a model for a prototypical contact site. AlphaFold predictions and *in vitro* binding assays identified a small region in the intermembrane space

**Funding:** This study was supported in part by project grant SNF 205200 to A.S. and by a grant of the NCCR RNA & Disease, a National Centre of Competence in Research (grant number 205601) to A.S both funded by the Swiss National Science Foundation (https://www.snf.ch/en). The funders had no role in study design, data collection and analysis, decision to publish, or preparation of the manuscript.

**Competing interests:** The authors have declared that no competing interests exist.

region of TAC60 that binds p166 forming contact sites. *In vivo* expression of various TAC60 and/or p166 mutants followed by immunoprecipitations demonstrates that contact site formation is driven by the hydrophobic effect and independent of the conserved charged amino acids present at the TAC60-p166 interface. The TAC is unique to Kinetoplastids, understanding the molecular architecture of the TAC60-p166 contact site could therefore inform the development of drugs that disrupt this critical interaction.

## Introduction

All organisms need to segregate their replicated genomes to their daughter cells during cell division. Within eukaryotes the same applies for mitochondria and plastids, which evolved from bacteria and have retained an own genome essential for their function [1]. The genome of mitochondria is organized in a number of discrete DNA-protein complexes, termed nucleoids, which in Opisthokonts such as mammals and fungi are distributed all over the organelle and associated with the mitochondrial inner membrane (IM) [2–4]. However, the molecular nature and exact architecture of the nucleoid-IM interactions is still unclear. This is different for the single mitochondrion of the parasitic protozoan *Trypanosoma brucei* and its relatives, which contains a single unit and highly concatenated genome termed kinetoplast DNA (kDNA) [5–7]. It consists of two genetic elements, maxicircles (22 kb, 35 copies each) and minicircles (1 kb, ca. 5000 copies each), which form a single large disk-shaped nucleoid. The kDNA is constitutively linked to the tripartite attachment complex (TAC) - a physical structure which extends across the IM and the outer membrane (OM) to the basal body (BB) of the single flagellum (Figs 1A and S1). The function of the TAC is to link the segregation of two single unit structures, the kDNA and the BB [5,8,9]. Thus, segregation of the replicated kDNAs is coupled to the segregation of the old and the new flagellum prior to cytokinesis. The single unit nature of the kDNA requires that its replication is coordinated with the nuclear cell cycle and BB segregation [10].

The highly unusual trypanosomal TAC can serve as a paradigm for a mitochondrial nucleoid that is constitutively attached to the IM and that extends to a cytoskeletal structure in the cytosol, the BB [11,12]. Intriguingly, the TAC has some resemblance to the mitotic spindles that segregate nuclear chromosomes in both open and closed mitosis [8]. The TAC and the spindle are both filament-based structures and extend, although in opposite directions, from the same type of microtubule (MT)-organizing centers: the BB (in case of the TAC) and the centriole (in case of the spindle). The BB and the centriole are homologous structures sharing many of the same subunits [13]. However, while the spindle filaments consist of MTs, the filaments of the TAC are much smaller, consisting of a single protein (p197) in the cytosol [14–16] and the protein pair (p166/TAC102) in the mitochondrial matrix [17–19]. The nuclear membrane-embedded spindle pole body in organisms showing closed mitosis serves as a platform to link the intranuclear spindle MTs to the cytosolic astral MTs [20]. A conceptually similar platform is formed by the four integral OM TAC subunits which link the cytosolic to the intramitochondrial TAC filaments [8].

The TAC consists of eight known essential subunits and can be subdivided into three molecular modules [8]. The outermost "cytosolic module" links the BB to the "OM module". It is made up of p197, a very large protein of approximately 670 kDa which contains approximately 26 tandem repeats of 175 aa in length [14–16]. The innermost "inner module" links the kDNA disk in the matrix to the IM. It comprises the kDNA-proximal TAC102 which interacts with the region corresponding to aa 71–210 of the α-helical p166 [17,19,21]. p166 forms

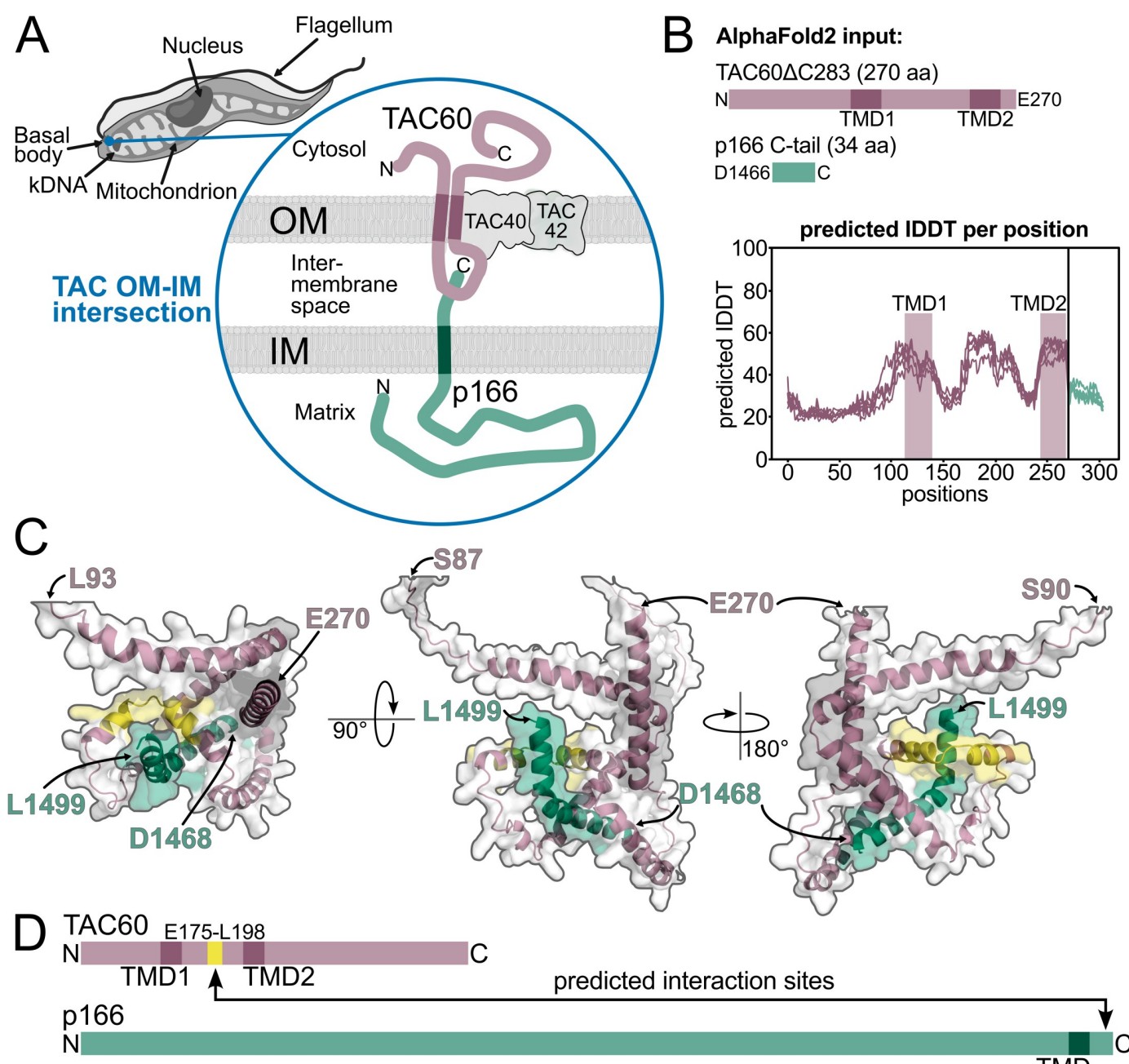

**Fig 1. AlphaFold2 predicts TAC60-p166 interaction. (A)** Model depicting the OM (outer membrane)-IM (inner membrane) contact site formed by TAC60 and p166 within the TAC (tripartite attachment complex). kDNA, kinetoplast DNA. **(B)** Depiction of the TAC60 (N-terminal 270 aa, purple) and p166 (C-terminal 34 aa, green) segments used as AlphaFold2 inputs. Structure prediction confidence (n = 5) is displayed as the predicted lDDT (local Distance Difference Test) per aa. TMD, transmembrane domain **(C)** Structural prediction of TAC60-p166 interactions from three angles. The following TAC60 regions are depicted in purple: L93-E270 (left), S87- E270 (middle), and S90-E270 (right). The TAC60 region E175-L198 predicted to fold into a kinked α-helix that binds to the C-terminus of p166 is highlighted in yellow. The C-terminal p166 segment D1466-L1499 is shown in green. **(D)** Schematic representation of the predicted interaction sites between TAC60 and p166 based on the AlphaFold2 model.

filaments in the matrix, is anchored in the IM via a single C-terminal transmembrane domain (TMD) and contains a 34 aa C-terminal region exposed to the intermembrane space (IMS) [17]. The central and most complex TAC subdomain is the OM module. It comprises the

peripheral OM protein TAC65 which interacts with both p197 of the cytosolic module and the integral OM protein pATOM36 [22,23] (S1 Fig). Intriguingly pATOM36, in addition of being an essential TAC subunit, is required for the biogenesis of a subset of α-helically anchored OM proteins [23,24]. The OM module contains three more integral membrane proteins: two beta barrel proteins, TAC40 and TAC42, as well as TAC60 [22,25]. The latter has two TMDs and its N- and C-termini face the cytosol [22]. The short IMS-exposed loop of TAC60 interacts with the C-terminus of p166 and thus connects the "OM module" with the "Inner module" [17].

The overarching principle of TAC biogenesis is a polar assembly of its subunits starting at the BB. Thus, depletion of a BB-proximal TAC subunit prevents assembly of all downstream TAC components [26]. How the 5 subunits of the "OM TAC module" are assembled is less clear. There is evidence that they form distinct assembly intermediates in the OM membrane independently of all other TAC subunits [8].

Here we have characterized the interaction between p166 and TAC60 on the molecular level. Using a combination of *in silico*, *in vitro*, and *in vivo* assays we identified which amino acids and structural features are critical for this interaction. The p166-TAC60 interaction is central for the understanding of TAC formation as it initiates the polar assembly of the TAC "inner module" that is guided by the "OM membrane module" [8]. Moreover, the p166-TAC60 interaction serves as a rare example of a permanent contact site between the IM and OM [27,28].

## Results

### AlphaFold2 predicts TAC60-p166 interaction

It has previously been shown that the N- and C-termini of the integral OM TAC subunit TAC60 are exposed to the cytosol, indicating that the sequence segment between the two TMDs (aa 142–237) must face the IMS [22]. Moreover, the only TAC subunit integral to the IM is p166. While most of p166 is exposed to the mitochondrial matrix, the protein has a single TMD near its C-terminus that is followed by a 34 aa C-terminal extension reaching into the IMS (Fig 1A and 1B). Immunoprecipitations have shown that this C-terminal extension is essential for the interaction with TAC60 and thus for cell growth [17]. To characterize the TAC60-p166 interaction in more detail the sequences corresponding to the C-terminal truncated variant of TAC60 (TAC60ΔC283), which was previously shown to be fully functional [22], and the C-terminal 34 aa of p166 (p166 C-tail) were used as inputs for an *in silico* analysis using the AlphaFold2 model (Fig 1B) [29, 30]. The confidence of the structure prediction (predicted local Distance Difference Test, IDDT) for TAC60 was rather mediocre between 20% and 60% (Fig 1B) and the two predicted TMDs did not align very well. However, a one-to-one interaction between the p166 C-tail and TAC60ΔC283 was predicted by the model (Fig 1C). The region interacting with the p166 C-tail corresponded to the TAC60 segment E175-L198. This was a plausible prediction as the interacting segment of TAC60 is located right in the center of the IMS loop (Fig 1D). Moreover, it also included the region where the structure of TAC60 was predicted with highest confidence. The AlphaFold2 model furthermore suggested that only the C-terminal half of the p166 C-tail might be involved in the TAC60 interaction.

### Microarray of TAC60 peptides defines p166 C-tail binding site

To confirm the AlphaFold2 prediction experimentally, an *in vitro* protein interaction study was performed using a TAC60 peptide microarray. To that end the p166 C-tail was recombinantly expressed in *E. coli* and purified by immobilized metal affinity chromatography using an N-terminal 6x His tag (S2 Fig). The purified p166 C-tail was directly labelled with a fluorochrome and incubated with the peptide microarray immobilized on a glass surface. The

microarray consisted of 179 overlapping 20 aa long purified synthetic peptides covering the entire TAC60 protein (Fig 2A). The resulting pattern of fluorescent signals was detected by a sensitive microarray scanning system and visualized by heat map analysis (Fig 2B). The heat map showed that the p166 C-tail could bind to five sets of peptides each covering a distinct region of TAC60. Two of these regions corresponded to TAC60 domains that are exposed to the cytosol. Moreover, the C-terminal one locates to a region that is dispensable for TAC function [22]. Two further hotspots for p166 C-tail binding are within, or overlap with, the TMDs. Thus, these four p166 C-tail binding sites cannot be physiologically relevant as *in vivo* they are not accessible for binding to the IMS-exposed C-tail of p166 (Fig 2B). However, one set of peptides that bound to the p166 C-tail, encompassing the TAC60 sequence Q178-M194, mapped to the center of the IMS-exposed loop of TAC60. This is essentially the same region of TAC60 (E175-L198) that was predicted to bind to the p166 C-tail according to the AlphaFold2 analysis (Fig 1C) and is in line with the known topologies of TAC60 and p166 (Fig 1A).

## TAC60-p166 interacting regions are conserved within Kinetoplastids

Homologues of TAC subunits, as the TAC itself, are exclusively found within the Kinetoplastids. S3 Fig shows a plot depicting the Shannon's entropy, a measure for the divergence of each position, of a multiple sequence alignment of TAC60 and p166 orthologues from a phylogenetically broad and balanced selection of 12 and 11 Kinetoplastid species, respectively. (*B. saltans* was excluded from the p166 alignment as its orthologue could not be confidentially identified in this species). Low Shannon's entropy values correspond to a high homology, whereas high values indicate high degree of divergence. Overall, the two proteins are only moderately conserved in the different species. However, the *T. brucei* TAC60 region (E175-L198) and the p166 C-tail region (D1466-L1499) which based on structure predictions and biochemical

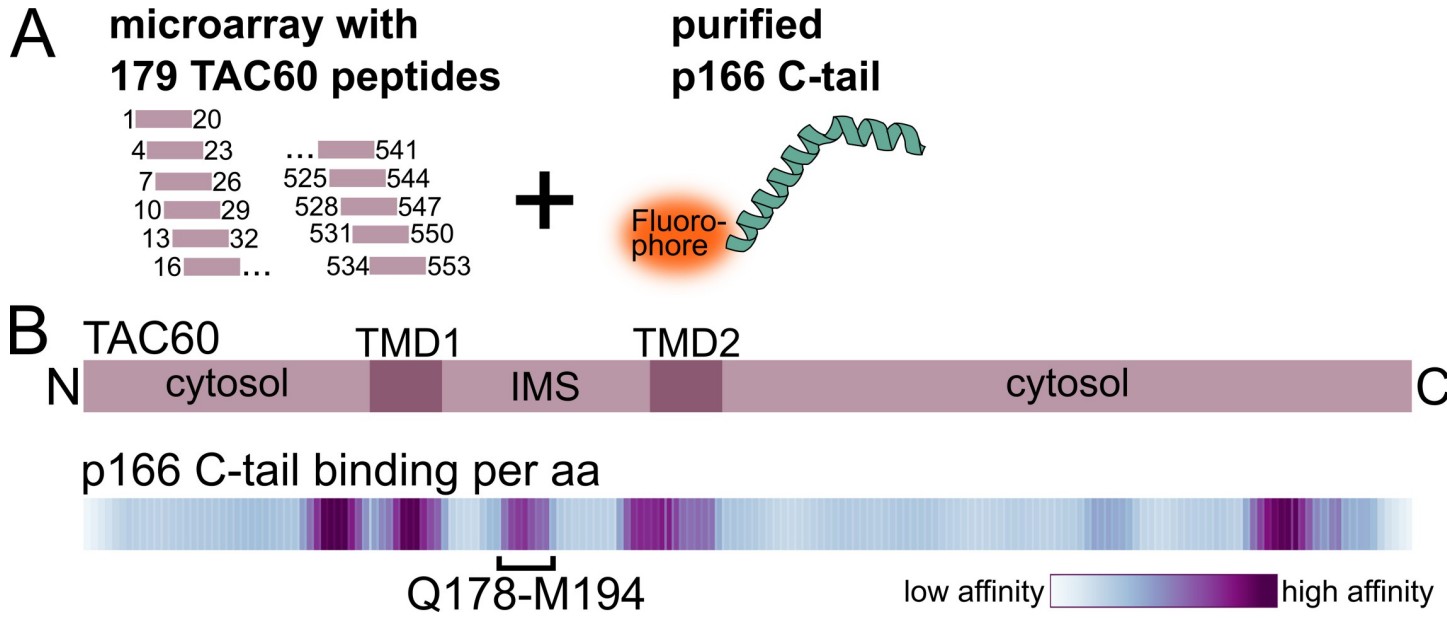

**Fig 2. Microarray of TAC60 peptides defines p166 C-tail binding site. (A)** Microarray setup for the *in vitro* peptide-protein interaction screen: 179 overlapping 20 aa long peptides of TAC60 were immobilized on a microarray. Interaction with the fluorophore-linked recombinant p166 C-tail was quantified. (For details and purification of the p166 C-tail see S2 Fig). **(B)** Top, domain structure of TAC60. TMD, transmembrane domain; IMS, intermembrane space. Bottom, corresponding heat map depicting the binding affinity of the p166 C-tail protein towards TAC60 peptides. The TAC60 segment Q178-M194 marks the minimal p166 binding site in the IMS domain of TAC60, which is in agreement with the domain identified in the AlphaFold2 structure model.

methods (Figs 1C and 2B) interact with each other are highly conserved (S3 Fig, yellow shading). The same is the case for the TMDs of both proteins (S3 Fig, grey shading).

In a next step we zoomed into the putative TAC60 and p166 interacting region of *T. brucei* and compared it with other Kinetoplastid species. The sequence logo in Fig 3A depicts the *T. brucei* TAC60 region (E175-L198) together with the corresponding region of TAC60 orthologues of the same species listed in S3A Fig. The analyzed TAC60 region contains four invariant residues: basic R181, the helix breaker P185, acidic E189, and hydrophobic L196. Moreover, positions 177, 180, 184, 188 and 192 are in all species occupied by hydrophobic amino acids. Thus, the *T. brucei* TAC60 segment interacting with the p166 C-tail is highly conserved across all analyzed Kinetoplastids. In addition, the AlphaFold2 model (Fig 1C) together with helical

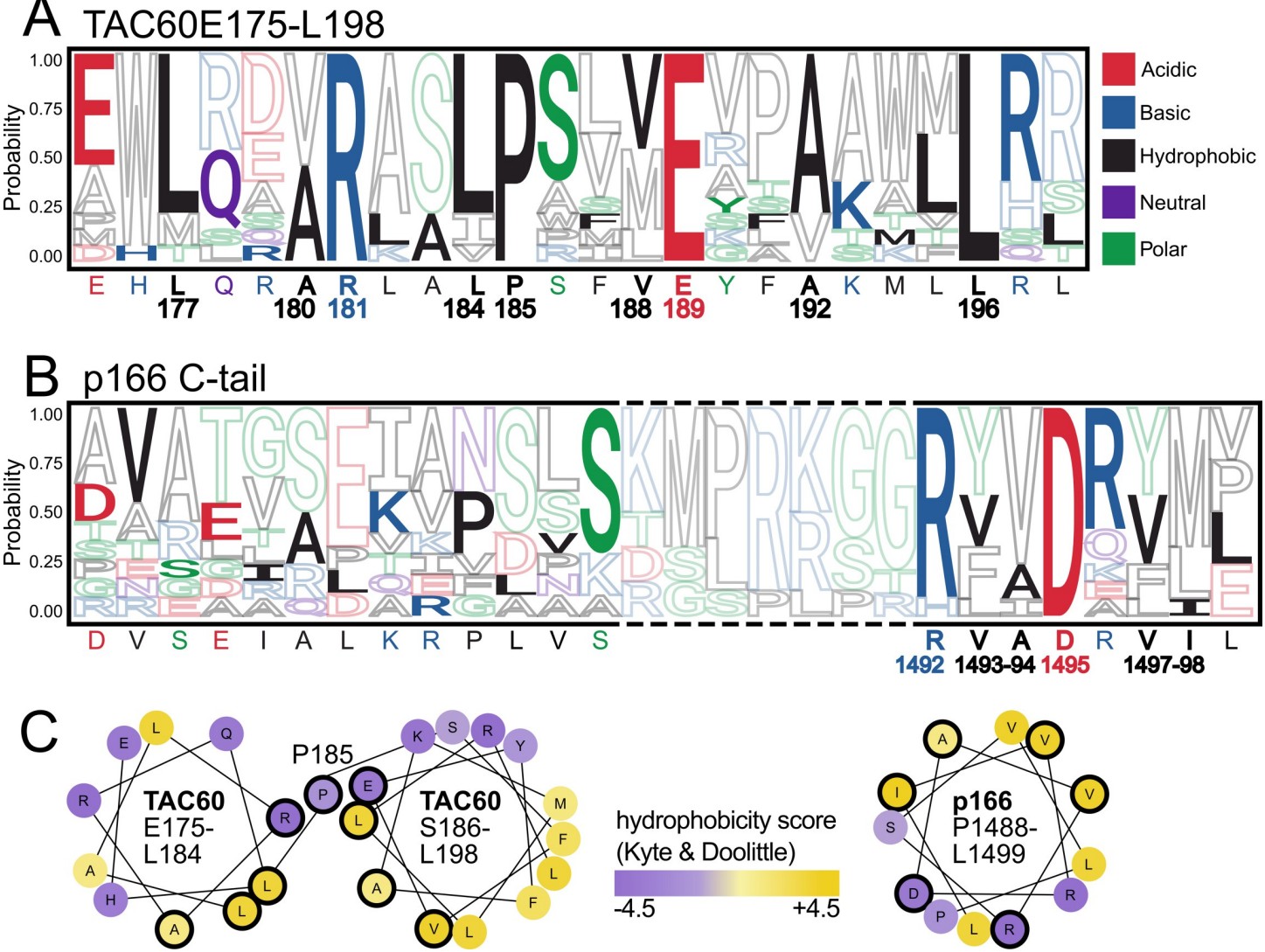

**Fig 3. TAC60-p166 interacting regions are conserved within Kinetoplastids. (A)** Sequence logo of a multiple sequence alignment (MSA) depicting the *T. brucei* TAC60 region (E175-L198) together with the corresponding region of TAC60 orthologues of 12 Kinetoplastid species (S3A Fig). The *T. brucei* sequence is depicted in filled letters and indicated at the bottom of the logo. Numbers refer to the *T. brucei* TAC60. **(B)** as in (A) but an MSA depicting the *T. brucei* p166 region D1479-L1499 is shown. Broken lines indicate a region containing insertion and or deletions in some Kinetoplastid species. **(C)** Helical wheel projections of the *T. brucei* TAC60 segments E175-L184 (left) and S186-L198 (middle) that are connected by P185. Helical wheel projection of the *T. brucei* p166 segment P1488-L1499 (right). Conserved aa and aa from conserved hydrophobic positions are encircled in black. Hydrophobicity is indicated from violet (hydrophilic) to yellow (hydrophobic).

wheel projection analyses (Fig 3C) suggest that a feature shared by all Kinetoplastid TAC60 proteins is that this region folds into two short amphipathic α-helices that are separated by the invariant P185.

The p166 C-tail sequence logo in Fig 3B depicts the *T. brucei* p166 C-tail region (D1479-L1499) aligned with the C-tail regions of p166 orthologues of the same species analyzed in S3B Fig. Overall the p166 C-tail region is less conserved than the TAC60 segment it interacts with. Note that the alignment contains some gaps due to small insertions and deletions relative to the *T. brucei* sequence (S4 Fig). The region mostly affected is indicated by broken lines in Fig 3B. However, the C-terminal 8 aa of *T. brucei* p166 (except for the last one which is absent in some species) are conserved (S4 Fig). This sequence contains a conserved basic amino acid (mostly R) at position 1492 and an invariant D1495. Moreover, positions 1493/1494/1497 and 1498 contain exclusively hydrophobic amino acids in all Kinetoplastids. Similar to what was observed for TAC60, AlphaFold2 and helical wheel analyzes suggest that the C-terminal 12 aa of *T. brucei* p166 fold into a short α-helix, the amphiphilic nature of which is conserved in all Kinetoplastids (Fig 3C).

### *In vivo* system to monitor TAC60-p166 interactions

Which features of the TAC60 region (E175-L198) and the p166 C-tail region (D1466-L1499) are important for their mutual interaction? To find out we devised an *in vivo* system allowing pulldown experiments to test whether mutations in the binding domains of either of the two proteins interfere with the TAC60-p166 C-tail interaction.

The system is based on a tetracycline-inducible RNAi cell line that targets the TAC60 mRNA region (nucleotides 1220–1629) encoding the C-terminal part of the protein (Fig 4A and 4B). Note that the efficiency of the RNAi was monitored in cell lines that in addition to expressing the mutant TAC60 variants also expressed an *in situ* tagged endogenous allele of TAC60 carrying a C-terminal myc-tag (insets in left panels of Fig 4B and 4C). However, growth curves and immunoprecipitations were done in transgenic cell lines having two wild-type alleles of TAC60.

It has previously been shown that tetracycline-inducible, ectopic expression of TAC60 variants results in the essentially complete replacement of the endogenous TAC60 by the ectopically expressed variant [22] (Fig 4C, inset left panel). The TAC60 variant lacking the C-terminal 283 aa and carrying a C-terminal 3x myc-tag, termed TAC60ΔC283-myc, was used as a positive control in our assay. Due to the C-terminal truncation it was not affected by the RNAi. Immunofluorescence analysis of isolated flagella which are still connected to the TAC [25] shows that TAC60ΔC283-myc can be fully integrated into the TAC (Fig 4C, right panel). Moreover, expression of TAC60ΔC283-myc fully complemented the growth inhibition observed in the TAC60 RNAi cell line (Fig 4C, left panel) [22].

Pulldown experiments with TAC subunits are challenging because the fully assembled TAC is insoluble in non-ionic detergents [17]. The OM TAC subunits are an exception because a small fraction of these proteins is found in detergent-soluble subcomplexes representing assembly intermediates [8,22] (Fig 4D). This is different for the full length IM TAC subunit p166 which is essentially insoluble [17]. Thus, we transfected, a C-terminally HA-tagged mini-version of p166 (142 aa in length) which lacks the N-terminal 1357 amino acids but includes the TMD and the IMS-exposed C-tail (mini-p166-HA) into the TAC60 RNAi cell line. To ensure that mini p166-HA was imported into mitochondria it was N-terminally fused to the mitochondrial targeting sequence of trypanosomal mtHsp60 (Fig 4A) [17]. The mini-p166-HA was correctly integrated into the IM, interacted with the IMS domain of TAC60 and

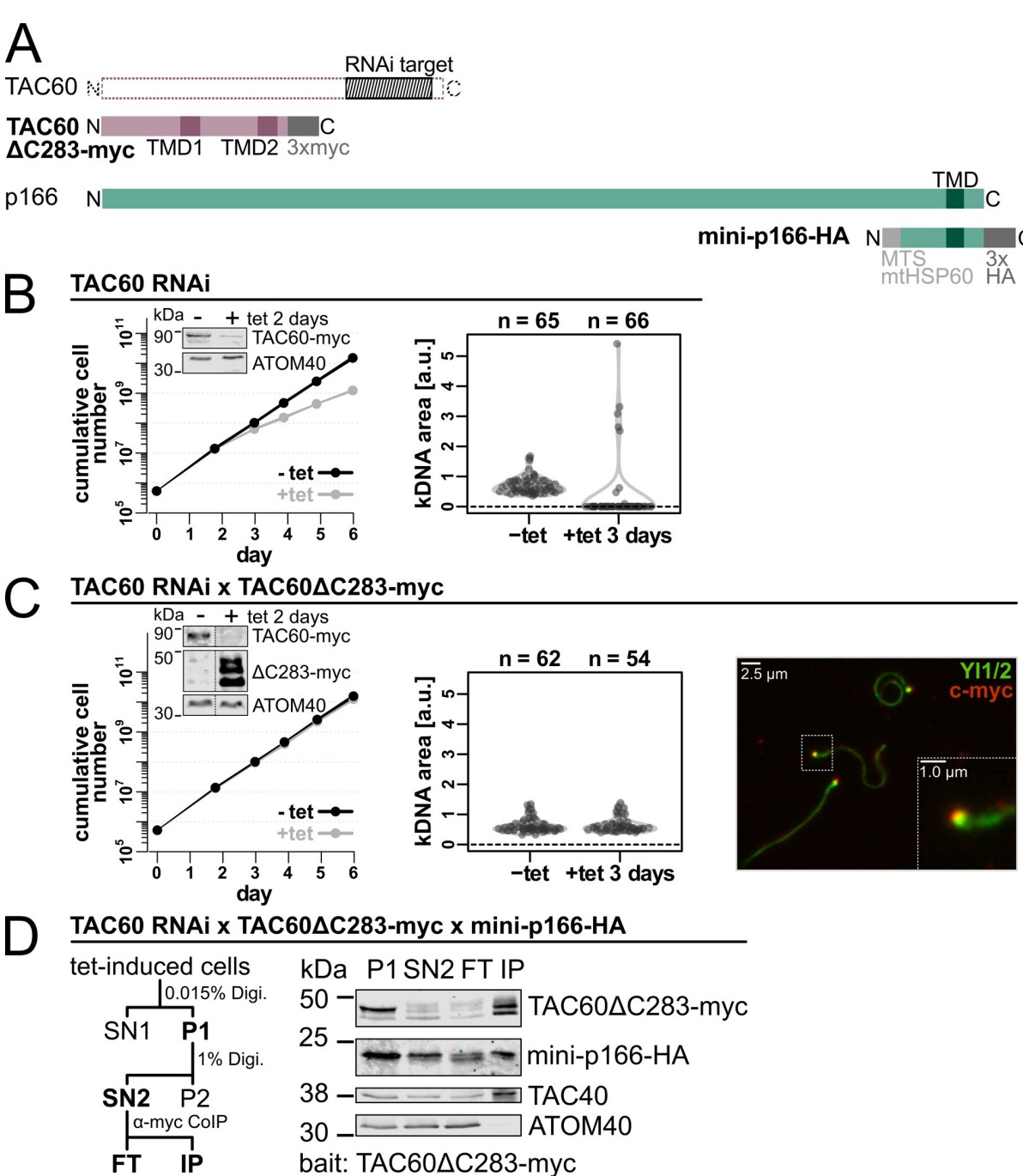

**Fig 4. *In vivo* system to monitor TAC60-p166 interactions. (A)** Schematic representations of TAC60ΔC283-myc and mini-p166-HA compared to the corresponding full length proteins. The region targeted by the TAC60 RNAi is shaded in grey and mapped to the full length protein. TMD, transmembrane domain; MTS, mitochondrial targeting sequence **(B)** Left, growth of uninduced (-tet) and induced (+tet) TAC60-RNAi cell line of procyclic *T. brucei*. The linewidth reflects the mean +/- the standard deviation of n = 3 experiments. Inset: Immunoblot of *in situ* 3x myc tagged full length TAC60 levels in uninduced (-) and tet-induced (+) cells. Note that one of the endogenous alleles of TAC60 was in situ tagged with a C-terminal 3x myc-tag to monitor the efficiency of the RNAi. ATOM40 serves as a loading control. Right, combined violin and sina diagrams of DAPI-stained kDNA area measurements, indicated as arbitrary units (a. u.), in the uninduced and induced TAC60 RNAi-cell line. Numbers of analyzed cells are indicated at the top. A kDNA area value of zero means the complete loss of the kDNA. **(C)** As in (A) but a TAC60-RNAi cell line complemented by TAC60ΔC283-myc is shown. Inset as in (A) but expression of TAC60ΔC283-myc is also monitored. Right, Immunofluorescence of extracted flagella of the same cell line probed for TAC60ΔC283-myc (red) shows the TAC60 variant gets integrated into the TAC. Tyrosinated tubulin and TbRP2, detected by YL1/2 (green) serves a marker for the flagellum and basal body. **(D)** Left, workflow of the digitonin (Digi.)- based cell fractionation assay used for pulldown experiments. Right, immunoblot of the pulldown experiment. A cell line shown in (C) induced for two days that also expresses mini-p166-HA was analyzed. P, pellet; SN, supernatant; FT, flow through; IP, eluate of immunoprecipitation. TAC40 and ATOM40 serve as positive and negative controls, respectively.

was fully detergent-soluble [17]. However, due the large N-terminal truncation which prevents its interaction with TAC102 and thus the kDNA, it was not functional [17].

Fig 4D shows a pulldown assay in which TAC60ΔC283-myc was used as a bait. TAC60ΔC283-myc is fully functional and serves as a positive control for all tested TAC60 variants. A crude mitochondrial fraction, the pellet (P1) of 0.015% digitonin extracted cells, was further extracted with 1% digitonin, resulting in a supernatant termed SN2. Whereas the fully assembled TAC remained insoluble and was recovered in the P1 pellet, a small fraction of the TAC60ΔC283-myc variants and TAC40, which likely represent assembly intermediates, were solubilized by this treatment. Throughout our study we consistently observe multiple bands for the various TAC60 variants. This is likely mainly due to as yet unknown posttranslational modifications, see [22] for a more detailed discussion. Mini-p166-HA, in contrast to the TAC60ΔC283-myc variants, was essentially completely recovered into the SN2 fraction. Subsequently SN2 was incubated with anti-myc-beads and processed for pulldown (Fig 4D). The result showed that TAC60ΔC283-myc together with the mini-version of p166-HA was recovered in the bound fraction (IP) indicating they interact with each other. TAC40 was also found in the IP fraction because it binds to TAC60 independent of its interaction with p166 and thus serves as a positive control. ATOM40, the integral OM pore subunit of the protein translocase, does not interact with the TAC and serves as a negative control (Fig 4D).

Finally, it has been previously observed that expression of mini p166 in the presence of the endogenous full length p166 caused a slight growth phenotype and a decrease of cells containing normal kDNA [17]. This weak dominant negative effect could be explained because mini-p166-HA likely competes for localization with the wild-type p166 [17]. Thus, to monitor the putative effect of TAC60ΔC283-myc mutants on the kDNA segregation process, they were also expressed in the TAC60-RNAi cell line that did not express the mini-p166-HA. Impairment of TAC function and thus kDNA segregation resulted mainly in kDNA loss. Moreover, over-replication of kDNAs in the few cells that have retained the kDNA was also observed [9] (Fig 4B, right panel and 4C, middle panel).

## A kinked helix in TAC60 is necessary but not sufficient for p166 binding

In the first TAC60 mutant tested, termed TAC60-nohelix, we replaced the segment Q178-M194 encompassing the minimal p166-binding site predicted by the TAC60 peptide binding array (Fig 2B). It was exchanged with the peptide (SALQMELIEPTPHILIP) of the same length, which contains three prolines and is predicted to be unable to form an α-helix. The result shows that while the TAC60-nohelix mutant still interacts with the OM TAC subunit TAC40, it cannot pull down mini-p166 (Fig 5A). This was expected considering the entire minimal p166-binding site was replaced and suggests that an α-helical structure of the binding site in TAC60 might be required for the interaction.

Indeed, two short α-helices separated by the invariant P185 is a feature of the *T. brucei* TAC60 p166-binding site that is highly conserved in all Kinetoplastids (Fig 3A). We therefore produced a TAC60 mutant, termed TAC60-P185E, in which the invariant P185 was replaced by an E. This resulting sequence is predicted to form a single α-helix covering the entire TAC60 p166-binding region. Intriguingly, the TAC60-P185E mutant lost the capability to interact with mini-p166 (Fig 5B) suggesting that an α-helix with a kink in the center is required for TAC60-p166 interaction.

In the last mutant of this series, termed TAC60-p197helix, P185 was left unchanged. However, the two short α-helices flanking P185 were replaced by peptides of the same length that were modelled after the α-helical repeat region of the previously characterized TAC subunit p197 [15]. The resulting p166 binding region in the TAC60-p197helix was predicted to fold

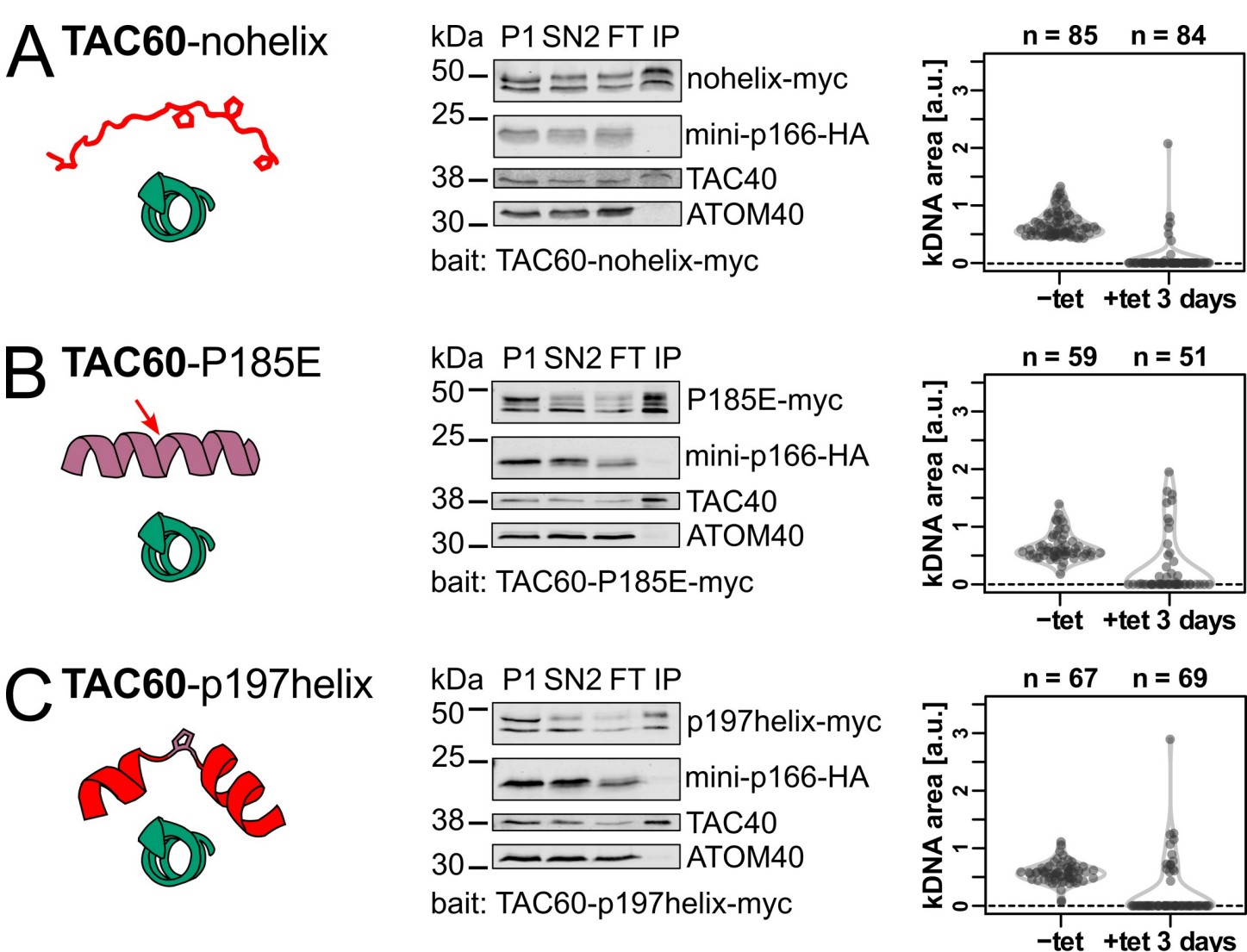

**Fig 5. A kinked helix in TAC60 is necessary but not sufficient for p166 binding. (A)** Left, schematic structural depiction of the TAC60-p166 interaction for the TAC60-nohelix mutant. Replaced residues of TAC60 are shown in red, the unmodified p166 C-tail is shown in green. Middle, immunoblot analysis of a pulldown experiment of the two days tet-induced TAC60-RNAi cell line complemented by the TAC60-nohelix mutant that also expresses mini-p166-HA. P, pellet; SN, supernatant; FT, flow through; IP, eluate of immunoprecipitation. TAC40 and ATOM40 serve as positive and negative controls, respectively. Right, combined violin and sina diagrams of DAPI-stained kDNA area measurements, indicated as arbitrary units (a. u.), of a tet-induced cell line for TAC60-RNAi and TAC60-nohelix expression. Numbers of analyzed cells are indicated at the top. A kDNA area value of zero means the complete loss of the kDNA. **(B)** and **(C)** as in (A) but the TAC60-P185E and TAC60-p197helix mutants were analyzed. The mutated residues are depicted by the red arrow (B) or by the red helix segments (C).

into a kinked α-helix, just as the corresponding wildtype sequence, but has different biochemical properties. The results in Fig 5C show that the TAC60-p197helix mutant was not able to interact with mini-p166, suggesting that a kinked α-helix, while necessary, is not sufficient to mediate binding of TAC60 to p166.

Finally, the panels on the right in Fig 5 show that, as would be expected, exclusive expression of all three TAC60 mutants interfered with kDNA segregation.

## Conserved charged amino acids are dispensable for TAC60-p166 interaction

The best-conserved amino acids in the p166-interacting TAC60 α-helix are R181 and E189 which flank the helix-breaking P185. R181 and E189 face the same side of the kinked α-helix and thus may face p166. For the TAC60-interacting α-helix of p166 the best conserved amino acids are the closely spaced R1492 and D1495 located on the same side of the p166 α-helix. Considering the highly conserved nature of the two pairs of charged amino acids in TAC60 and p166, suggests that the TAC60-p166 interactions might be mediated by ionic bonds between opposite charges of R181 (in TAC60) and D1495 (in p166) as well as E198 (in TAC60) and R1492 (in p166), respectively (Fig 6A).

The importance of ionic bonds for TAC60-p166 interactions was experimentally tested by three TAC60 variants in which either R181 and/or E189 were mutated. In the first mutant, termed TAC60-R181A/E189A, both R181 and E189 were each replaced by an uncharged A (Fig 6B, left). In the second mutant, termed TAC60 R181E/E189R, the opposite charges were switched (Fig 6C, left), and in the third mutant, termed TAC60-E189R, E189 was switched to R resulting in a p166-TAC60 interacting kinked α-helix that contains two positive charges (Fig 6D, left). The results of the TAC60 pulldown experiments showed that all three TAC60 mutants still interacted with mini-p166 (Fig 6B, 6C and 6D, middle panels). In line with these results we did not observe impairment of kDNA segregation in cell lines that exclusively express the three mutant TAC60 proteins (Fig 6B, 6C and 6D, right panels).

These experiments were complemented with two p166 mutants, termed p166-R1492A and p166-D1495A, in which either R1492 or D1495 were replaced by a neutral A (Fig 6E and 6F, left). Moreover, a third mutant, termed p166-R1492D/D1495R, was also tested in which R1492 and D1495 were switched (Fig 6G, left). The results showed that in all TAC60 pulldown experiments the mutant p166 versions were recovered in the bound fraction (Fig 6E, 6F and 6G, middle panels).

Our results show that based on the *in vivo* binding assay all positively or negatively charged amino acids, R181/E189 in TAC60 or D1495/R1492 in p166, even though they are highly conserved, are dispensable for the mutual interaction of the two proteins. This excludes that the interaction between TAC60 and p166 is due to ionic bonds.

## TAC60-p166 interaction depends on conserved hydrophobic amino acids

Charged amino acids are not required for the TAC60-p166 interaction and the interaction between the two proteins is maintained in the TAC60 R181E/E189R mutant. This strongly suggests that the side of the *T. brucei* p166 α-helix containing hydrophobic amino acids faces the kinked TAC60 α-helix, rather than the side with the highly conserved D1492 and E1495 (Fig 7A, top). Thus, we tested whether the interaction between TAC60 and p166 requires the presence of hydrophobic amino acids.

Positions 177, 180, 184, 188, 192 and L196 in TAC60 of *T. brucei* are all occupied by hydrophobic amino acids which are oriented to the same side of the kinked TAC60 α-helix, a feature that is highly conserved in all Kinetoplastids (Figs 1C and 3A). Thus, we expressed two TAC60 mutants, in which either all six positions (termed TAC60-fullφ) (Fig 7B) or all except positions L177 and L196 (termed TAC60-4φ) (Fig 7C, left) were replaced by the structurally most similar hydrophilic amino acid (Fig 7A, bottom). The results of the pulldown experiments showed that both the TAC60-fullφ and the TAC60-4φ mutants cannot pull down mini-p166 indicating that at least some of the four amino acids mutated in TAC60-4φ are essential for the TAC60-p166 interaction (Fig 7B and 7C, middle panels). In line with these results kDNA

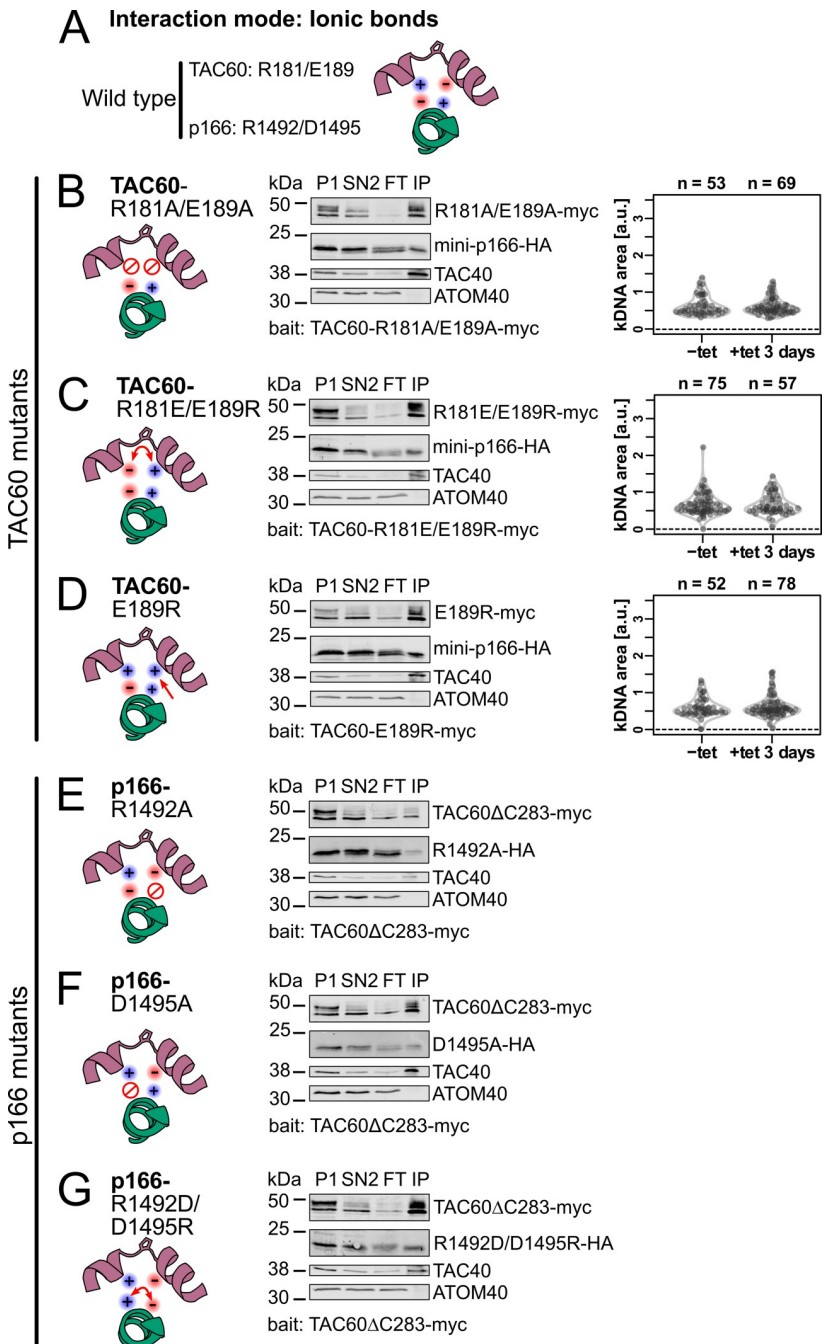

**Fig 6. Conserved charged amino acids are dispensable for TAC60-p166 interaction. (A)** Schematic structural prediction of the model where TAC60 (purple) interacts with p166 (green) via ionic bonds. **(B), (C)** and **(D)** Left, schematic structural depiction of the TAC60-p166 interaction for the indicated TAC60 mutants. Middle, immunoblot analyses of pulldown experiments of the two days tet-induced TAC60-RNAi cell lines complemented by the indicated TAC60 mutants that also express mini-p166-HA. P, pellet; SN, supernatant; FT, flow through; IP, eluate of immunoprecipitation. TAC40 and ATOM40 serve as positive and negative controls, respectively. Right, combined violin and sina diagrams of DAPI-stained kDNA area measurements, indicated as arbitrary units (a. u.), of the same cell lines but without mini-p166-HA expression. Numbers of analyzed cells are indicated at the top. A kDNA area value of zero means the complete loss of the kDNA. **(E), (F)** and **(G)** as above but TAC60-RNAi cell lines complemented by TAC60ΔC283-myc and the indicated mini-p166-HA mutants were analyzed.

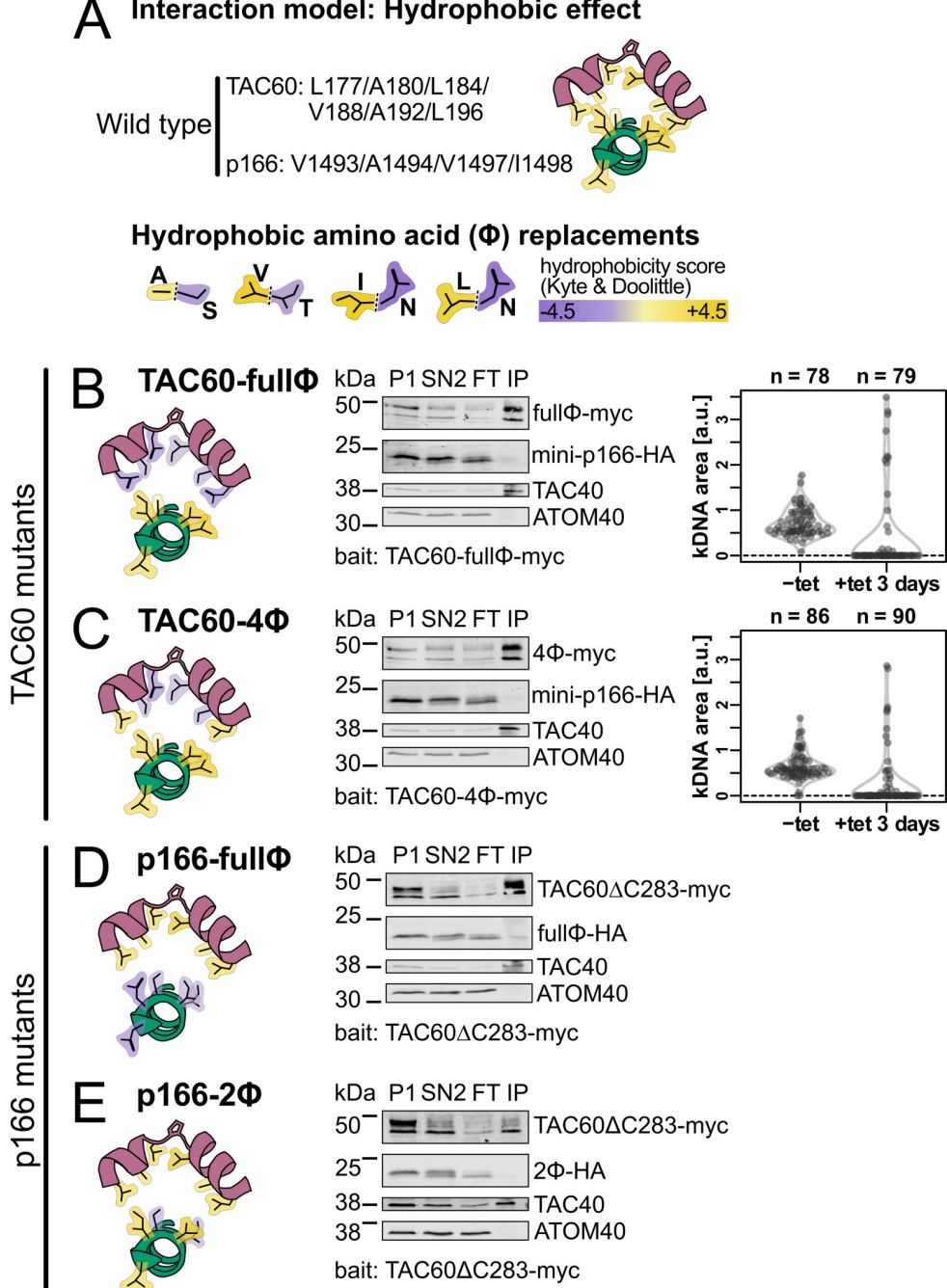

**Fig 7. TAC60-p166 interaction depends on conserved hydrophobic amino acids.** **(A)** Top, schematic structural prediction of the model where the TAC60 (purple)-p166 (green) interaction is driven by the hydrophobic effect. Hydrophobic residues are depicted in yellow. Bottom, hydrophobic aa were replaced by the structurally most related hydrophilic aa, as indicated. **(B)** and **(C)** Left, schematic structural depiction of the TAC60-p166 interaction for the indicated TAC60 mutants. Middle, immunoblot analyses of pulldown experiments of the two days tet-induced TAC60-RNAi cell lines complemented by the indicated TAC60 mutants that also express mini-p166-HA. P, pellet; SN, supernatant; FT, flow through; IP, eluate of immunoprecipitation. TAC40 and ATOM40 serve as positive and negative controls, respectively. Right, combined violin and sina diagrams of DAPI-stained kDNA area measurements, indicated as arbitrary units (a. u.), of the same cell lines but without mini-p166-HA expression. Numbers of analyzed cells are indicated at the top. A kDNA area value of zero means the complete loss of the kDNA. **(D)** and **(E)** as above but TAC60-RNAi cell lines complemented by TAC60ΔC283-myc and the indicated mini-p166-HA mutants were analyzed.

segregation was impaired in cell lines that exclusively expressed the mutant TAC60 proteins (Fig 7B and 7C, right panels).

The *T. brucei* TAC60 binding region of p166 contains seven hydrophobic amino acids (L1489, V1490, V1493, A1494, V1497, I1498, L1499) all of which are predicted to face the kinked α-helix of TAC60. To test the importance of these amino acids for the TAC60-p166 interaction we produced two mutant mini-p166 variants. In the first one, termed p166-fullϕ (Fig 7D, left), all hydrophobic amino acids were replaced by their most similar hydrophilic counterparts, whereas in the second one, termed p166-2ϕ (Fig 7D, left), only the central V1493 and A1494 were replaced. The presence of hydrophobic amino acids at these two positions is conserved in all Kinetoplastids (Fig 3B). Note that position 1493 and 1497 may be occupied by a Y in some species which is ambiguously classified as either hydrophobic or polar, respectively. The results showed that in TAC60 pulldown experiments both p166-fullΦ and p166-2Φ were not recovered in the bound fraction (Fig 7D and 7E, right panels).

An independent confirmation that the TAC60-p166 interaction is mainly due to hydrophobic rather than ionic interaction is the fact that in a TAC60 pulldown experiment mini-p166 is still recovered in the eluate even in the presence of 0.75 M NaCl which is expected to interfere with electrostatic interactions (Fig 8).

In summary, these results show that the interaction between TAC60 and p166 depends on a kinked α-helix in TAC60, which has a hydrophobic surface on the inside of the bent region, that interacts with the hydrophobic side of the C-terminal α-helix of p166.

## Discussion

A detailed knowledge of protein-protein interactions is crucial for the understanding of cellular architecture and for gaining mechanistic insights into biological processes. In this study we have characterized the interaction between the two mitochondrial integral membrane proteins TAC60 (OM) and p166 (IM) of *T. brucei* at the molecular level. The two proteins are subunits of the TAC, which mediates the segregation of the duplicated single-unit mitochondrial nucleoids during the coordinated cell and mitochondrial division in trypanosomes and related organisms [10].

Using a combination of *in silico*, *in vitro*, and *in vivo* analyzes our results suggest a model of how the two proteins interact. For TAC60, the p166-binding site corresponds to the short E175-L198 segment. It contains three conserved charged amino acids and six highly conserved hydrophobic amino acids, forming two short amphiphilic α-helices that are separated by the

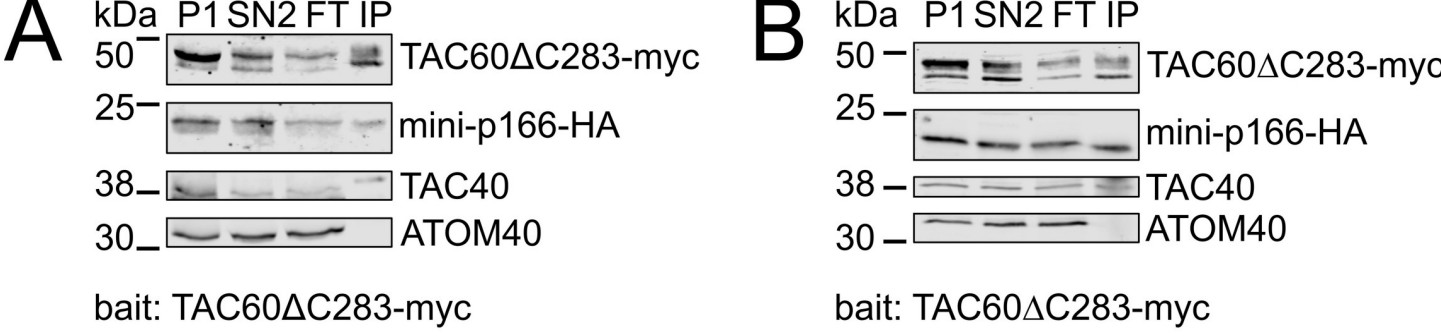

**Fig 8. TAC60-p166 interaction is salt resistant.** Immunoblot analyses of pulldown experiments of the two days tet-induced TAC60-RNAi cell line complemented by TAC60ΔC283-myc performed in the presence of 500 (**A**) and 750 (**B**) mM NaCl. P, pellet; SN, supernatant; FT, flow through; IP, eluate of immunoprecipitation. TAC40 and ATOM40 serve as positive and negative controls, respectively.

P185. The hydrophobic face of each α-helix lines the inside of the bent region, creating a hydrophobic pocket. For p166, the TAC60 binding site corresponds to the C-terminal 11 amino acids of p166. It contains two conserved charged amino acids and four highly conserved hydrophobic amino acids and forms an amphiphilic α-helix. The hydrophobic side of the p166 α-helix faces the hydrophobic pocket formed by the TAC60 E175-L198 segment.

Three key mutants prevented the interaction between TAC60 and p166, supporting the suggested model: i) The TAC60-P185E mutant, which cannot form a kink between the two short α-helices (Fig 5B); ii) The TAC60-4ϕ mutant, in which four conserved hydrophobic amino acids on the inside of the hydrophobic pocket were replaced by hydrophilic ones (Fig 7C), and iii) The p166-2ϕ mutant in which two conserved hydrophobic amino acids on the TAC60-facing side of the p166 α-helix were replaced by hydrophilic ones (Fig 7E). Further corroborating this model, replacing or switching the highly conserved charged amino acids in either TAC60 or p166 in various combinations did not affect TAC60-p166 interactions (Fig 6). This indicates that the conserved charged amino acids are oriented diametrically opposed to the hydrophobic sides of the TAC60 and the p166 α-helices that form the binding interface. We conclude that the TAC60-p166 interaction depends on hydrophobic amino acids at conserved positions in both the TAC60 and the p166 α-helices. Thus, the interaction is ultimately driven by the tendency to minimize the exposure of hydrophobic side chains to water, which is known as the hydrophobic effect. Structural predictions of the TAC60-p166 interaction interface of distantly related Kinetoplastids shows that the model proposed for *T. brucei* likely applies for the whole group (S5 Fig).

If the charged amino acids do not contribute to the TAC60-p166 interaction why are they so highly conserved (Fig 3)? Presently we cannot answer this question. It is possible that they modulate the efficiency of mitochondrial protein import, and that in the cell lines in which the charged amino acids were mutated an import phenotype would be masked because the ectopically expressed mutant TAC60 subunits were overexpressed. Alternatively, as all our experiments were done in procyclic cells, it could be that the charged amino acids and/or the amphiphilic nature of the α-helices have a function in other life cycle stages.

The specific role of TAC60 and p166 within the TAC is to form contact sites between the mitochondrial OM and IM. While the TAC is specific for Kinetoplastids, OM-IM contact sites are an essential feature of mitochondria in all eukaryotes. The most prominent examples are the ones associated with protein translocases [28] and with the mitochondrial contact site and cristae organizing system (MICOS) [31,32].

The translocase of the OM (TOM) complex transiently interacts with the translocase of the inner membrane 23 (TIM23) complex. This interaction is based on a complicated interaction network between the IMS domains of Tom22/Tom40/Tom7 and Tim50/Tim23/Tim21, respectively, and is greatly stabilized by precursor proteins that are being translocated [33,34]. Transient OM-IM contact sites are also formed by the OM protein voltage dependent anion channel (VDAC) which binds to both hydrophobic precursor proteins that are bound to small TIM chaperones in the IMS and to the carrier translocase, the TIM22 complex [35]. In both examples the function of the OM-IM contact sites is to facilitate the transfer of precursor proteins from the TOM complex to the respective protein translocase in the IM.

The hetero-oligomeric MICOS complex forms the cristae junction, a narrow passage in the IM that links the inner boundary membrane, which runs parallel to the OM, with invaginations in the IM termed cristae [31,32]. The MICOS subunit Mic60 is anchored in the IM. It has a large domain exposed into the IMS which is associated with Mic19. The Mic60/Mic19 module forms contact sites with the OM, by interacting with variety of OM proteins including the sorting and assembly machinery (SAM) and the TOM complex, VDAC, and the GTPase Miro [36,37].

Even though OM-IM contact sites are widespread and functionally important, not much is known about the underlying molecular interactions required for their formation and maintenance. The TAC is specific for Kinetoplastids, but it can serve as an example for a prototypical OM-IM contact site. The TAC is a single unit, permanent structure which is precisely localized opposite the single flagellum to which it connects the kDNA. This is different to the OM-IM contact sites described above which are often transient and/or highly dynamic and therefore more challenging to study [38]. The architectural unit of the TAC can be thought of as a cable connecting the BB to the kDNA (S1 Fig) [8]. The cytosolic part of each cable is formed by a single filamentous p197 molecule, which is plugged into the TAC OM module via TAC65 and extends further through other OM module subunits to TAC60. The IMS side of TAC60 then interacts with a single p166 molecule, which extends across the IM, forming a matrix filament that connects to the kDNA. The exact stoichiometry of the TAC subunits within the structure remains unknown. However, the TAC subunits can easily be detected by immunofluorescence. This indicates that the TAC must have a highly repetitive structure consisting of several hundreds of the described TAC cables arranged in a parallel manner. This simplifies *in vivo* studies of the TAC OM-IM contact site because interfering with the TAC60-p166 interaction prevents the formation of each individual TAC cable and thus of the whole structure.

OM-IM contact sites can have multiple functions. The contact sites found in MICOS, besides maintaining the mitochondrial architecture, have been implicated in phospholipid transport and metabolism, protein import, and signaling pathways [36, 37]. However, while the TAC is essential in both procyclic and in bloodstream forms of trypanosomes, its function is restricted to kDNA segregation [26]. The evidence for this comes from the observation that the TAC is dispensable for normal growth of the L262P bloodstream form cell line, which due to a mutation in the γ-subunit of the ATP synthase, can grow in the absence of the kDNA [39]. This is perhaps surprising because for an exclusive tethering function, the OM TAC module consisting of four essential integral OM membrane proteins, appears to be unnecessarily complex.

In contrast to most other mitochondrial OM-IM contact sites, the trypanosomal TAC is a permanent structure. What could be the explanation for this? The kDNA disk, the flagellum and the BB are single unit structures in non-dividing cells and the BB is the master organizer of cellular architecture [10]. Proper BB duplication, maturation, segregation, and its positioning within the cell, ensures correct segregation of flagella, the replicated kDNAs and other structures during cytokinesis [40]. Thus, proper kDNA segregation in the cell is achieved, i) by coordinating kDNA replication with BB duplication and maturation, and ii) by the evolution of a physical tether that permanently hardwires the kDNA to the BB. In short, because the overarching principle of kDNA segregation is "physically coupled co-segregation with basal bodies" [12], this necessitates a permanent OM-IM contact site formed by TAC60 and p166.

The TAC60-p166 contact site must not only be temporally stable but also physically robust because the TAC makes the connection between two huge structures: the kDNA and the BB with the flagellum. Moreover, during mitochondrial fission and cytokinesis the old and the newly formed kDNA-TAC-BB supercomplexes are segregated within the highly viscous matrix and cytosol, respectively. Hence, the TAC60-p166 interaction must be strong enough, and the number of single unit TAC cables high enough, to bear this load.

Our study defined the binding interface of the OM-IM contact site formed by TAC60-p166 interaction in the trypanosomal TAC. Furthermore, we have shown that the formation and maintenance of the contact site does not rely on electrostatic interactions but is driven by the hydrophobic effect. Even though it is presently beyond the scope of our study, this knowledge may in the future allow to engineer synthetic OM-IM contact sites in organisms other than trypanosomes and thus open new ways to manipulate mitochondrial architecture.

Finally, knowing the critical features of the TAC60-p166 contact sites may help to find compounds that interfere with their formation. Because TAC60 and p166 are conserved in Kinetoplastids (Figs 3 and S4) but absent in mammals, such substances may form the basis of new drugs to combat the diseases that are caused by *T. brucei* and its relatives.

## Material and methods

### Protein structure predictions

Protein and protein complex structures were predicted using the AlphaFold2 model [29], implemented in ColabFold [30,41], and visualized with PyMol (version 2.5, Schrödinger, LLC). Input sequences for the predicted structures in Fig 1C were TAC60 (Tb427.07.1400, aa 1–270) and p166 (Tb427tmp.02.0800, aa 1466–1499). Predictions were made using the pdb100 template data base. Default settings were used where applicable and all structures were relaxed using amber.

### Protein purification

The coding sequence of a 6x His tag-3x myc tag was cloned upstream of the sequence coding for the C-terminal 34 aa of p166 by PCR and the coding sequence of the resulting fusion protein was inserted into an *E. coli* expression vector derived from the "parallel" expression vector family [42]. *E. coli* BL21 [43] was transformed with the resulting construct and expression of the fusion protein was induced for 3 hours by 1 mM isopropyl-β-D-thiogalactopyranosid (IPTG). Cells were harvested by centrifugation at 5'000 g and washed with phosphate-buffered saline (PBS) (137 mM NaCl, 2.7 mM KCl, 10 mM $Na_2HPO_4$, and 1.8 mM $KH_2PO_4$, pH 7.4) before cell lysis. Cell lysis was done using a high pressure homogenizer (HPL6, Maximator GmbH) in cell lysis buffer (20 mM 3-(N-morpholino) propanesulfonic acid (MOPS) at pH 7.5 containing 150 mM NaCl, 10% glycerol, 10 mM imidazole) supplemented with 1 mM phenyl-methanesulfonyl fluoride (PMSF). Cell debris was removed by centrifugation (20 minutes, 13'000 g at 4˚C). The resulting supernatant was incubated with Ni-NTA agarose beads (Cube Biotech) for 3 hours at 4˚C. Beads were washed in cell lysis buffer supplemented with 40 mM imidazole, and cell lysis buffer supplemented with 200 mM imidazole was used for elution of the fusion protein. The final eluate was concentrated using a 3 kDa size filter (Millipore) to approximately 1 μg/μl as determined by the BCA test (Thermo Scientific) [44].

### In vitro peptide-protein interaction screening

The *in vitro* peptide-protein interaction assay was conducted by JPT Peptide Technologies GmbH (Germany). TAC60 peptides 20 aa in length with a 3 aa shift relative to the preceding peptides covering the whole TAC60 sequence were synthesized and immobilized on microarray slides yielding a total of 179 different peptides. The purified p166 fusion protein was labelled using the DyLight microscale antibody labeling kit (Thermo Scientific). The immobilized TAC60 peptides were incubated with 10, 1, 0.1, 0.01 and 0.001 μg/ml of the labelled fusion protein for 1 hour at 30˚C. Bound proteins were detected by fluorescence emitted after excitation with a high-resolution laser scanner at 635 nm. The resulting signals for all peptides were extrapolated to scores for each aa of the TAC60 sequence by summing up the scores of all overlapping peptide regions (Fig 3B).

### Multiple sequence alignments and helical wheel projections

Sequences of TAC60 and p166 orthologs of Kinetoplastids were obtained from the TriTrypDB database [45]. The following orthologs were used for the analyses: *T. brucei* (TAC60:

Tb427.07.1400, p166: Tb427tmp.02.0800), *T. cruzi* (TAC60: TcCLB.508209.30, p166: TcCLB.509589.40), *Angomonas deanei* (TAC60: ADEAN_000333400, p166: ADEAN_000718300), *Blechomonas ayalai* (TAC60: Baya_042_0220, p166: Baya_075_0150), *Bodo saltans* (TAC60: BSAL_45895), *Crithidia fasciculata* (TAC60: CFAC1_290011300, p166: CFAC1_220045700), *Endotrypanum monterogeii* (TAC60: EMOLV88_260009700, p166: EMOLV88_130019600), *Leishmania aethiopica* (TAC60: LAEL147_000431100, p166: LAEL147_000188600), *L. amazonensis* (TAC60: LAMA_000507700, p166: LAMA_000208900), *L. donovani* (TAC60: LdBPK_260530.1, p166: LdBPK_131340.1), *L. major* (TAC60: LmjF.26.0560, p166: LmjF.13.1600), *Leptomonas pyrrhocoris* (TAC60: LpyrH10_01_8410, p166: LpyrH10_20_0130), *Paratrypanosoma confusum* (TAC60: PCON_0005540, p166: PCON_0032280). Multiple sequence alignments (msa) were calculated with the R (version 4.3.2) package msa (version 1.36.1) [46] from Bioconductor (version 3.17) and visualized as sequence logos (Fig 3A and 3B) or as an msa (S4 Fig) using the ggseqlogo (version 0.2) [47] and ggmsa (Bioconductor, version 1.10.0) [47] packages, respectively. Shannon's entropy [48] for msa (S3 Fig) was calculated only for positions with <50% gaps using the base two logarithm version. Phylogenetic trees (S3 Fig) were created using the neighbour joining algorithm [49] implemented in the R package ape (version 5.8) [49] with distance tables calculated by the SeqinR package (version 4.2–36) [50]. Helical wheel projections (Fig 3C) were created with the helixvis package (version 1.0.1) [51] in R. Hydrophobicity values were taken from Kyte and Doolittle [52].

## Transgenic cell lines

All cell lines derive from a single marker *T. brucei* 427 cell line [15] grown at 27°C in SDM-79 supplemented with 5% (v/v) fetal calf serum. Tetracycline-inducible RNAi of TAC60 (Tb927.7.1400/Tb427.07.1400) targets nucleotides 1220–1629 of the open reading frame [22]. To monitor TAC60 RNAi efficiency in Figs 4B, 4C and S6, one of the alleles of TAC60 was C-terminally tagged with 3x myc using a PCR construct based on plasmids of the pMOtag series [53]. C-terminally 3x myc-tagged TAC60 mutants are based on a C-terminally truncated TAC60 version termed TAC60ΔC283 [22] and were cloned into modified pLEW100 vectors [54] for tetracycline-inducible expression. S1 Table summarizes all TAC60 mutants. The C-terminally 3x HA-tagged N-terminally truncated version of p166 (Tb927.11.3290/Tb427tmp.02.0800) was previously described as "mini-p166-HA" [17]. A full list of all mutants of mini-p166 is given in S1 Table. Note, the protein lengths of Tb927.11.3290 and Tb427tmp.02.0800 differ by two aa (Tb927.11.3290 Q998 and S999 are missing in Tb427tmp.02.0800), we therefore exclusively refer to residue numbers of the Tb427tmp.02.0800 aa sequence. The control experiments that the TAC60 mutants used in our study completely replace the endogenous TAC60 and are fully integrated into the TAC are provided in S6 and S7 Figs, respectively.

## Flagella extraction and immunofluorescence microscopy

Flagella extraction was performed as described previously [55]. In summary, two days tetracycline-induced cell cultures were supplemented with EDTA (5 mM final) before harvesting. Harvested cells were lysed on ice for 10 minutes with extraction buffer (10mM $NaH_2PO_4$, 150mM NaCl, 1mM $MgCl_2$, pH 7.2) supplemented with 0.5% (v/v) Triton X-100 and centrifuged (3'000 g, 3 minutes, 4°C). The resulting cytoskeletal pellet was resuspended in extraction buffer without Triton X-100 and centrifuged as above. The resulting pellet was resuspended in 200 µl containing $10^8$ cell equivalents extraction buffer containing 1 mM $CaCl_2$, incubated on ice for 30 minutes and centrifuged (10'000 g, 10 minutes, 4°C). Finally the extracted flagellar

pellet was washed twice with PBS and resuspended in PBS at $10^7$ cell equivalents/50 μl. Subsequently, the flagellar fraction was processed for immunofluorescence microscopy. 50 μl of isolated flagella were allowed to settle on glass slides before fixation with 4% paraformaldehyde for 10 minutes. Fixed flagella were washed with PBS, chilled on ice and blocked with PBS containing 2% (w/v) bovine serum albumin (BSA) before incubation with two rounds of primary (anti myc, YL1/2) and the corresponding secondary antibodies diluted in PBS containing 2% BSA. For more information about the antibodies used see below. Pictures were acquired on a DMI6000B microscope equipped with a DFC360 FX monochrome camera and LAS X software (Leica Microsystems). Images were processed using Fiji software.

## kDNA area quantification

kDNA area measurements were performed on three days tetracycline-induced cells. Cells were harvested by a low spin centrifugation and after washing with PBS allowed to settle on glass slides. Settled cells were fixed with 4% paraformaldehyde and Vectashield, containing 4′,6-diamidino-2-phenylindole (DAPI) for DNA visualization, was added to the final samples before mounting the cover slides. Z-stack images were acquired and projected to one plane using Fiji software. All images were processed with identical contrast settings and were converted to binary files for kDNA area quantification. kDNA area was only measured when the entire cell was visible and where the kDNA was in the focal plane. The absence of the kDNA was determined by eye. Violin [56] and sina [57] graphs were generated using the R packages vioplot (version 0.5.0, https://github.com/TomKellyGenetics/vioplot) and SinaPlot (version 1.1.0).

## Immunoprecipitations

The TAC60-myc pulldown experiments were performed as described previously [17]. Solubilized mitochondria-enriched fractions of two days tetracycline-induced cells were generated by a two-step digitonin extraction. First, washed cells were lysed in SoTE buffer (20 mM Tris HCl pH 7.5, 0.6 M sorbitol, 2 mM ethylenediaminetetraacetic acid (EDTA)) containing 0.015% (w/v) digitonin and 1x cOmplete, Mini, EDTA-free protease-inhibitor-cocktail (Roche). Following a centrifugation step (6'700 g, 5 minutes, 4°C) the mitochondria-enriched pellet fraction (P1) was resuspended in a 20 mM Tris HCl pH 7.4, 100 mM NaCl, 10% glycerol, 0.1 mM EDTA, containing 1% (w/v) digitonin and 1x of cOmplete protease-inhibitor-cocktail as above. After another centrifugation (20'000 g, 15 minutes, 4°C) the resulting supernatant (SN2) was processed for immunoprecipitations and incubated with anti-c myc beads (Sigma) for 2 hours at 4°C. Subsequently the beads were washed in the same buffer containing only 0.1% (w/v) digitonin. For elution the beads were boiled in SDS-PAGE sample loading buffer lacking β-mercaptoethanol. Samples were collected for the P1, SN2, FT, and IP fractions (Fig 4D) and $5 \times 10^6$ (P1, SN2, and FT) or $5 \times 10^7$ (IP) cell equivalents were used for SDS-PAGE and subsequent immunoblot analysis.

## Antibodies

The following non-commercial antibodies were used. The dilutions of the antibodies are indicated in parentheses for immunoblots (IB) and immunofluorescence (IF) analyses, respectively. The polyclonal rabbit antisera against TAC40 (Tb927.4.1610) (IB 1:100) and ATOM40 (Tb927.9.9660) (IB 1:10'000) were described before in [15] and [58], respectively. The monoclonal rat antibody YL1/2 (IF 1:1'000) that recognizes tyrosinated α-tubulin [59] and the basal body protein TbRP2 [60] was a gift from Keith Gull. Commercially available antibodies were used as follows: monoclonal mouse anti-myc antibody (Invitrogen, 132500; IB 1:2'000, IF

1:50), monoclonal mouse anti-HA antibody (Enzo Life Sciences AG, CO-MMS-101 R-1000; IB 1:5'000).

Secondary antibodies for immunoblot analyses were IRDye 680LT goat anti-mouse and IRDye 800CW goat anti-rabbit (both from LI-COR Biosciences; IB 1:20'000). Secondary antibodies for immunofluorescence microscopy were goat anti-mouse Alexa Fluor 596, goat anti-rat Alexa Fluor 488 (both from Thermo Scientific; IF 1:1'000).

## Supporting information

**S1 Fig. Architectural unit of the tripartite attachment complex (TAC).** Single unit TAC cable connecting the basal body to the kinetoplast DNA (kDNA). The three molecular TAC modules and the individual TAC subunits are indicated. The TAC consists of several hundreds of TAC cables arranged in a parallel manner. OM, outer membrane; IM, inner membrane. (TIF)

**S2 Fig. His-tag affinity purification of the p166 C-tail. (A)** Depiction of the intermembrane space exposed C-terminus of p166 (D1466-L1499) that was N-terminally fused to 6x His and 3x myc tags and recombinantly expressed in *E. coli*. **(B)** Workflow for the His-tag affinity purification of the fusionprotein. **(C)** Left, Ponceau S stain of a blotted SDS-gel monitoring the purification of the recombinant fusion protein. Right, immunoblot staining of the purified protein fraction (Reten.) using an anti-myc antiserum. (TIF)

**S3 Fig. TAC60 and p166 are conserved in Kinetoplastids. (A)** Multiple sequence alignment of TAC60 orthologues from the Kinetoplastid species shown in the phylogenetic tree on the left was analyzed using a Shannon's entropy plot. **(B)** Multiple sequence alignment of p166 orthologues from the Kinetoplastid species shown in the phylogenetic tree on the left was analyzed using a Shannon's entropy plot. The right graph shows a magnification of the C-terminal p166 region depicted by the dashed red line. (TIF)

**S4 Fig. Multiple sequence alignment of p166.** Sequence alignment of the C-terminal region of p166 orthologues of the same Kinetoplastid species as were used for Fig 3B. (TIF)

**S5 Fig. AlphaFold2 predictions of the TAC60-p166 interaction for diverse kinetoplastid species.** Models depicting the predicted TAC60- p166 interaction interface in *T. brucei*, *T. cruzi*, *L. donovani*, and *A. deanei*. The predicted structure for *T. brucei* is identical to the model shown in Fig 1. For the predictions in the other species the following input sequence segments were used: *T. cruzi* TAC60 (1–233 aa), p166 (1349–1384 aa); *L. donovani* TAC60 (1–312 aa), p166 (1160–1204 aa); *A. deanei* TAC60 (1–239 aa), p166 (986–1017 aa). The models display the conserved kinked α-helix of TAC60 beginning 10 aa upstream and ending 12 aa downstream of the conserved P (see Fig 3). The sidechains of the conserved hydrophobic aa are shown as sticks. While the predicted local Distance Difference Test (IDDT) scores were low for all predictions, a hydrophobic pocket is predicted for all the interaction interfaces. (TIF)

**S6 Fig. Expression of TAC60 mutants in the TAC60-RNAi cell line replaces the endogenous TAC60 with its mutated counterparts. (A)** Immunoblots showing that TAC60 RNAi results in efficient depletion of the endogenous *in situ* tagged TAC60-myc and the exclusive expression of the ectopically expressed TAC60ΔC283-myc variants analyzed in Fig 4. (B) and (C) as in (A) but TAC60ΔC283-myc mutants of Figs 6 and 7 were analyzed. ATOM40 serves

as a loading control.
(TIF)

**S7 Fig. Immunofluorescence analysis of isolated flagella with TAC60 mutants.** Immunofluorescence images of extracted flagella of the indicated mutant TAC60 cell lines show that the mutant proteins (red) co-fractionate with flagella and colocalize with or very close to the basal body. This indicates that the mutant proteins are integrated into the TAC. Tyrosinated tubulin and TbRP2, detected by YL1/2 (green) serves a marker for the flagellum and basal body. Broken line mark original and enlarged insets.
(TIF)

**S1 Table. Summary of TAC60 and p166 variants.** The control experiments that the TAC60 mutants used in our study completely replace the endogenous TAC60 and are fully integrated into the TAC are provided in S6 and S7 Figs, respectively
(PDF)

**S1 Data. Numerical data for all graphs presented in the study.**
(XLSX)

**S1 Raw images. Original images for all gels and blots.**
(PDF)

## Acknowledgments

We thank Elke Horn for excellent technical assistance.

## Author Contributions

**Conceptualization:** Philip Stettler, André Schneider.

**Formal analysis:** Philip Stettler, Salome Aeschlimann.

**Funding acquisition:** André Schneider.

**Investigation:** Philip Stettler, Bernd Schimanski, Salome Aeschlimann.

**Methodology:** Philip Stettler.

**Supervision:** Bernd Schimanski, André Schneider.

**Validation:** Philip Stettler.

**Visualization:** Philip Stettler, André Schneider.

**Writing – original draft:** Philip Stettler, André Schneider.

**Writing – review & editing:** Philip Stettler, Bernd Schimanski, Salome Aeschlimann, André Schneider.

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
