## [Decision Letter · Decision Letter 0]

29 Oct 2024

PPATHOGENS-D-24-02128Molecular characterization of the permanent outer-inner membrane contact site of the mitochondrial genome segregation complex in trypanosomesPLOS Pathogens Dear Dr. Schneider, Thank you for submitting your manuscript to PLOS Pathogens. Your manuscript was fully evaluated at the editorial level and by three independent reviewers.  All three reviewers expressed their appreciation for the well-written narrative commending the rigorous experimental design and the high quality of the data. The reviewers also acknowledged the importance of the findings on the molecular interactions of the TAC components. While this study is primarily concerned with the inheritance of mitochondrial DNA (mtDNA) in trypanosomes and their relatives, the reviews agreed on the significance of in-depth analysis of  the molecular architecture of the permanent contact sites, which allows the conclusions to be extended beyond the field of kinetoplastid biology.

As can be seen from the peer review, all three reviewers had formal requirements that I asked you to fulfill. After a comprehensive review of the second reviewer's comments, I believe that while it would be interesting to perform the proposed experiments to determine the binding affinity of recombinant p166 C-tail and the various TAC60 mutants, or to analyze the TAC complexes using native gels, these steps are not necessary. Reviewer 3 suggests that in silico analysis of the binding capacity of TAC60 and p166 proteins from related species (e.g. T. cruzi and Leishmania spp.) can be performed to complement Fig. S3/S4. This suggestion might be worth considering. Should the proposed modifications to the text and figures be implemented, it is my intention to reach a final decision without necessitating a second round of peer review.   Please submit your revised manuscript within 30 days Dec 28 2024 11:59PM. If you will need more time than this to complete your revisions, please reply to this message or contact the journal office at plospathogens@plos.org. Please include the following items when submitting your revised manuscript:*
A rebuttal letter that responds to each point raised by the editor and reviewer(s). You should upload this letter as a separate file labeled 'Response to Reviewers'. This file does not need to include responses to any formatting updates and technical items listed in the 'Journal Requirements' section below.*
A marked-up copy of your manuscript that highlights changes made to the original version. You should upload this as a separate file labeled 'Revised Manuscript with Track Changes'.*
An unmarked version of your revised paper without tracked changes. You should upload this as a separate file labeled 'Manuscript'. If you would like to make changes to your financial disclosure, competing interests statement, or data availability statement, please make these updates within the submission form at the time of resubmission. Guidelines for resubmitting your figure files are available below the reviewer comments at the end of this letter. We look forward to receiving your revised manuscript. Kind regards, Alena Zíková, PhDGuest EditorPLOS Pathogens Dominique Soldati-FavreSection EditorPLOS Pathogens Michael Malim

Editor-in-Chief

PLOS Pathogens

orcid.org/0000-0002-7699-2064 **Journal Requirements:** **Additional Editor Comments (if provided):****Reviewers' Comments:** Reviewer's Responses to Questions

**Part I - Summary**

Reviewer #1: Stettler et al. conducted a comprehensive investigation into the molecular interactions between the outer membrane protein TAC60 and the inner membrane protein p166 within the tripartite attachment complex (TAC) of Trypanosoma brucei. The TAC is a crucial structure that links the basal body of the flagellum to the kinetoplast, playing a key role in the proper segregation of mitochondrial DNA during cell division in this protozoan parasite.

By employing a combination of computational predictions and experimental approaches—both in vitro and in vivo—the authors identified specific protein regions and amino acids essential for the interaction between p166 and TAC60. Their analysis narrowed down the critical elements to hydrophobic side-chain contacts between a kinked α-helix in TAC60 and the C-terminal α-helix of p166. This detailed insight into the protein-protein interface advances our understanding of the structural basis for TAC function.

The study not only extends existing knowledge about the TAC but also provides important molecular details of the interaction between its components. The high-quality data, derived from rigorous experimental design and execution, lend strong support to the authors' conclusions. The narrative is well-written and effectively communicates the most salient points, guiding the reader through the complex methodologies and findings with clarity.

While the objective of the study is somewhat narrowly focused, its implications are significant. By elucidating the specific contacts critical for TAC assembly and function, the research offers valuable information that could inform future studies on kinetoplast segregation and flagellar attachment. This could have broader impacts on our understanding of mitochondrial inheritance and the biology of Trypanosoma brucei.

In conclusion, Stettler et al. have made a noteworthy contribution and their work enhances our understanding of the TAC and sets the stage for future investigations into the mechanisms of mitochondrial DNA segregation in eukaryotes.

Reviewer #2: Purpose of the study:

The authors aimed to investigate the molecular interface of the TAC60 and p166 interaction and its role in mitochondrial genome maintenance. Trypanosomes have a single copy mtDNA, known as kDNA, anchored to the basal body, via the essential tripartite attachment complex (TAC). This physical linkage enables co-segregation of the mtDNA along with the basal body during cell division. TAC is comprised of many multi-protein modules, with p197 linking the basal body to pATOM36 and TAC60 at the mitochondrial outer membrane. Previous studies have shown that p166, a mitochondrial matrix protein is essential for kDNA biogenesis (Zhao et al., 2008). Recently, a study from the authors group showed that p166 has a transmembrane domain, localizes to the IM and consists of a C-terminal tail that extends to the IMS, thereby linking the TAC to kDNA (Schimanski et al., 2022). However, the molecular architecture of the TAC60-p166 interaction at the IM reamins unclear.

Brief Summary:

In this study, the authors employ in silico, in vitro and in vivo methods to establish the minimum-binding region of TAC60 that interacts with p166. They validate AlphaFold predictions with in vitro peptide binding assay and phylogenetic analyses. In vivo pull down of TAC60 and its variants show that a helical kink in TAC60 and conserved hydrobhobic amino acids in the interface are necessary for the TAC60-p166 interaction and kDNA maintenance. The study provides significantly novel and very interesting data leading to detailed insights into a molecular understanding of the assembly of the mitochondrial genome segregation complex in trypanosomes.

The experiments are of high quality and appropriately controlled.

The manuscript can in principle be published as it is.

I have only a very few comments which the authors might find usefull to follow in a revised version of the manuscript:

Since the TAC60-p166 acts as a permanent physical linkage between the basal body and kDNA, it would be interesting to determine (or at least estimate) the binding affinity. The recombinant p166 C-tail from FigS2 could be tested for its binding with the various domains of TAC60 and mutants by isothermal calorimetry or similar methods. From their previous study (Schimanski et al., 2022), immunoprecipitation of mini-p166-HA can pulldown multiple TAC components. Hence, in vivo pulldowns cannot rule out indirect interactions. In vitro binding assays would provide more clarity to the strength of binding between TAC60 and p166.

It could be a good control to analyze the TAC complexes in mitochondria isolated from TAC60 and p166 mutant cells in comparison to the controls on Blue-Native PAGE gels whether the interactions that they observed in pulldowns form a native protein complex or how the native TAC complex behaves.

In Line 212, it should be specified Fig.4C, middle panel.

In Line 231, the authors state that Mini-p166-HA was completely recovered into the SN2 fraction referring to Fig. 4D, however there is a considerable amount of Mini-p166-HA present in the FT fraction, the text should be corrected as Mini-p166-HA was predominantly or mainly recovered into the SN2 fraction.

In Figure 4C, can the authors explain why three different bands are observed in ΔC283-myc immunoblot membrane?

In Figure 4 B and C, western blot analyses could be shown in separate panels instead of illustrating them on the graph as such small figures.

In Figures 4D and in Figures 5,6,7, in pulldown assays, TAC40 immunoblot membrane is cut quite close to the signal, the authors could cut the signal with more space around the edges, if it is possible.

In Figures 4B, 4C, 5A, 5B, 5C a.u. abbreviation should be described as arbitrary units in the figure legends as well as in other diagrams of DAPI stained kDNA area measurements.

In Figures 5A, 7E and 8A, in pulldown assays, TAC40 signal in the bound fraction (IP) runs higher than the other lanes. Can the authors explain the reason?

Reviewer #3: This study offers a detailed exploration of the molecular framework involved in mitochondrial genome segregation in Trypanosoma brucei, specifically focusing on the role of the tripartite attachment complex (TAC). The TAC serves as a physical link between the mitochondrial genome and the flagellum basal body, with two proteins, TAC60, located in the outer membrane, and p166, in the inner membrane, playing central roles. Through a combination of computational modelling, mutational analysis, in vitro and in vivo assays, the researchers mapped the interaction between these proteins.

One of the study’s key revelations is the identification of a short kinked α-helix in TAC60 that binds to the C-terminal α-helix of p166. Interestingly, this interaction is driven by hydrophobic forces rather than the more common electrostatic interactions, highlighting the importance of hydrophobic residues in maintaining this critical link. By pinpointing the minimal binding regions and emphasising the significance of hydrophobic interactions, the research deepens our understanding of how the TAC operates at a molecular level to ensure proper mitochondrial genome segregation. The study adds valuable insights into the structural organisation of the TAC and also opens avenues for further research into potential therapeutic targets within this excessively complex system.

The manuscript is mainly easy to read, well-written and clear, with good-quality data. It has a well-argued analysis of the problem and comprehensible and concise results. It is sometimes a little tricky to follow. It only requires one major modification and a few minor ones, and if done, in my opinion, it would make it acceptable for publication in PLoS Pathogens.

**Part II – Major Issues: Key Experiments Required for Acceptance**

Reviewer #1: None.

Reviewer #2: (No Response)

Reviewer #3: Major comments

An obvious question raised from reviewing this work is: can TAC60 proteins from other species bind to the w/t of T. brucei’s proteins P166? And, can p166 from other species bind to the w/t of T. brucei’s TAC60? It would be interesting to test this, focusing on genes from T. cruzi and a Leishmania species (closely or distantly related). Having asked the questions, I realize it is perhaps beyond the scope of this current work to do this in the lab, but in silico, it is possible. An alpha-fold2 analysis to address these two questions, albeit limited, would be relatively simple and easy to do (Similar to Fig1C).

**Part III – Minor Issues: Editorial and Data Presentation Modifications**

Reviewer #1: None.

Reviewer #2: (No Response)

Reviewer #3: Minor comments

1. Figure S6, in C, the alignment of the IFA signals on the TAC60 mutants is misaligned and needs to be checked. 

2. The authors used the monoclonal antibody YL1/2 to probe and locate basal bodies, and they stated that this is “Tyrosinated tubulin, detected by YL1/2”. YL1/2 indeed binds tyrosinated tubulin, but it has been shown that in trypanosome basal bodies, YL1/2 detects TbRP2, ref, (Volume 168, Issue 4, August 2017, Pages 452-466). For clarity, the authors should note this in their manuscript.

3. The methods section has page and alignment-related problems.

4. What happened is unclear, but the conversion to .pdf format may have separated hyphenated words throughout the text and removed the Greek symbol for alpha.

PLOS authors have the option to publish the peer review history of their article (what does this mean?). If published, this will include your full peer review and any attached files.

Reviewer #1: No

Reviewer #2: No

Reviewer #3: No

---

## [Editor Report · Decision Letter 1]

18 Nov 2024

Dear Prof Schneider,

We are pleased to inform you that your manuscript 'Molecular characterization of the permanent outer-inner membrane contact site of the mitochondrial genome segregation complex in trypanosomes' has been provisionally accepted for publication in PLOS Pathogens.

Best regards,

Alena Zíková, PhD

Guest Editor

PLOS Pathogens

Dominique Soldati-Favre

Section Editor

PLOS Pathogens

Michael Malim

Editor-in-Chief

PLOS Pathogens

orcid.org/0000-0002-7699-2064
---

## [Editor Report · Acceptance letter]

22 Nov 2024

Dear Prof Schneider,

We are delighted to inform you that your manuscript, "Molecular characterization of the permanent outer-inner membrane contact site of the mitochondrial genome segregation complex in trypanosomes," has been formally accepted for publication in PLOS Pathogens.

Best regards,

Michael Malim

Editor-in-Chief

PLOS Pathogens

orcid.org/0000-0002-7699-2064